# A Low-Cost Gamified Urban Planning Methodology Enhanced with Co-Creation and Participatory Approaches

Ioannis Kavouras [1,*,†], Emmanuel Sardis [1,†], Eftychios Protopapadakis [2,†], Ioannis Rallis [1,†], Anastasios Doulamis[1,†] and Nikolaos Doulamis[1,†]

1   School of Rural, Surveying and Geoinformatics Engineering, National Technical University of Athens, 10682 Athens, Greece
2   Department of Applied Informatics, School of Information Sciences, University of Macedonia, 54636 Thessaloniki, Greece
*   Correspondence: ikavouras@mail.ntual.gr
†   These authors contributed equally to this work.

**Abstract:** Targeted nature-based small-scale interventions is an approach commonly adopted by urban developers. The public acceptance of their implementation could be improved by participation, emphasizing residents or shopkeepers located close to the areas of interest. In this work, we propose a methodology that combines 3D technology, based on open data sources, user-generated content, 3D software and game engines for both minimizing the time and cost of the whole planning process and enhancing citizen participation. The proposed schemes are demonstrated in Piraeus (Greece) and Gladsaxe (Denmark). The core findings can be summarized as follows: (a) the time and cost are minimized by using online databases, (b) the gamification of the planning process enhances the decision making process and (c) the interactivity provided by the game engine inspired the participation of non-experts in the planning process (co-creation and co-evaluation), which decentralizes and democratizes the final planning solution.

**Keywords:** digital twin; urban planning; nature-based solution; gaming tools; game engines; free and open source; co-creation; co-evaluation

## 1. Introduction

Urban planning includes technical and political processes, focused on the design and development of land usage and the building environment. Such processes consider various environmental factors (e.g., water, air, etc.) and the infrastructures passing into and out of urban areas (e.g., transportation, communication network, etc.) for different human activities within it [1]. Consequently, urban planning procedures require an effective management of heterogeneous data sources (e.g., geo-spatial data, 3D building models, etc.).

The tremendous urbanization signifies the need to design smart and environmentally friendly cities, which aim at improving the health and well-beings of the citizens. The increased availability of spatial data from smart cities, data analytics, machine learning and artificial intelligence permits the creation of digital twins, which can be changed and updated, along with its physical equivalent [2]. A digital twin is a 3D representation (up to a specific degree of accuracy) of an existing place (e.g., village, city, island, etc.) [3] and can be used for urban planning [4]. Furthermore, it makes it easier for urban planners, citizens and stakeholders to cooperate in the planning process.

Multiple simulation and serious games, involving city management and planning, have been developed [5–7]. These games investigate the impact of multiple aspects over a long period of time [8]. Software and application development aspects range from critical thinking boosting [9] to emergent property management [10]. As such, it is considered

feasible and beneficial to pursue the utilization of the interactive environment (e.g., game engine world and digital twins) for urban planning processes.

A core component of the environment design, for urban planners and architects, is the availability of the intervention plans through an electronic format, where they will be capable of applying their thoughts and transformations. Three-dimensional visualization (digital twin) is becoming a necessity since it makes it easier for humans to see patterns and relationships among objects and allows planners to decide accordingly on how to proceed. Additionally, the 3D visualization provides an interactive and navigable environment, which can be shared by everyone and enhances the co-creativity between planners and non-planners.

In addition, 3D virtualization enables evaluation of the outcome of the interventions beforehand, without affecting activities and environmental conditions of life afterwards. Such digital developments require many tools and on-site visits from engineers to gather the necessary information (images, drone views, recordings, etc.); in a second phase they need to be transformed and implemented in a common digital 3D environment. As a last phase, the planners operate and work above this 3D visualization, by applying their solutions, which can be shared with non-expert groups for evaluation. Non-expert groups may send feedback to the planners (co-evaluation).

In this work, we propose a methodology that combines 3D reconstruction (digital twin) methodologies, using online free and open-source and/or user-generated data, with virtualization using game engine software for urban planning processing. The proposed methodology aims to minimize, as much as possible, the time and cost of the whole planning process and increase the co-creation and co-evaluation between expert groups and non-expert groups. This can be achieved by sharing the urban planning solution(s) between these groups, using online platforms or offline procedures (e.g., physical presentation, distribution via USB or CD, etc.).

The proposed scheme resulted in a low-cost, co-creative simulation game, which allows participants to plan and interact over the area(s) of interest. It is low-cost, because the whole planning process depends on free and open-source data, software and tools, which in most cases minimize the time needed to achieve the same result by using state of the art approaches (i.e., on-site visits, measurements, etc.). It is co-creative because multiple groups of people are cooperating and sharing their opinions either by providing ideas in the planning phase or by giving feedback and proposing alternative ideas during the evaluation phase. These groups can be experts such as the project's coordination team, other architect planners and engineers, or non-experts such as citizens, the city council, etc.

As an example, two case studies are presented in this manuscript, following our proposed methodology. The first case study is at Dilaveri Coast in the city of Piraeus (Greece). The second case study is at Pileparken 6, a social housing estate in Gladsaxe (Denmark). Both of these cases are characterized by a lack of green spaces; thus, they are a good example for urban planning using nature-based solutions. The co-creation and co-evaluation phases have not been tested in these case studies because they need additional technologies and preparation that cooperate with the ethics and laws of the country. This preparation needs time, which was not available when this manuscript was written.

The rest of this paper is organized as follows: Section 2 provides details related to urban planning methodologies. Section 3 provides a theoretical background concerning the technologies used in our proposed methodology. Section 4 presents the proposed methodology, including the adopted steps, software and tools needed. Section 5 describes the data collection and design stages of the proposed methodology. Section 6 describes the co-creation and co-evaluation approaches. Section 7 applies the proposed methodologies to the case studies and presents the results. Section 8 further discusses the methodology of this work, and Section 9 concludes the work.

## 2. Related Work

Multiple data sources, including the Internet of Things, virtual reality, augmented reality and mixed reality, along with the development of the metaverse [11], provide a solid ground for innovative methodologies creating digital twins. Digital twins can be used for simulating reality, as described in the work of [12], where the authors developed a prototype digital twin of Herrenberg, Germany, compromised of (a) a 3D model of the built environment, (b) a street network model, (c) an urban mobility simulation, (d) a wind flow simulation and (e) a number of empirical qualitative and quantitative data using volunteered geographic information. Thus, digital twins can be used as an alternative to the state of the art city needs, such as urban planning and design practices [13].

Urban planning, based on a digital twin model, is feasible and an example is provided by Schrotter et al., for the city of Zurich [14]. The innovation in their work is the usage of a virtual city model of Zurich (digital twin [15]) for intervention planning in the areas of environment (e.g., noise, air pollution and mobile phone radiation modelling), energy (e.g., solar potential analyses), urban planning (visualization of construction projects, shadow and visibility analyses) and in other areas. Additionally, they recreated Zurich in a video game called Minecraft for attracting younger people to participatory processes like co-creation [16].

Video games, game engines and tools have been used widely in urban planning procedures, especially as participatory tools. Indicative example include the work of Ampatzidou et al. [17], where the authors identified a research-practice gap in the current state of participatory urban planning practices in three European cities (Groningen, Vienna and Genk). Their developed tools were as follows: (a) complex urban issues were illustrated [18], (b) social learning was evoked [19] and (c) participation was made accessible to the general public [20]. Their tools were acknowledged by planners and policymakers; however, inexperienced users remained skeptical, while limiting their gamified applications within participatory urban planning practices.

A different approach suggests the creation of a game simulation. Some examples could be the works of Aguilar et al. [21] and John et al. [22]. The first work uses a serious game [23], named Metropolis, for smart city planning, as well as for promoting e-participation [24] tools within the context of urban planning. The second work evaluates the effectiveness of the simulation game SimCity™4 for educating students in undergraduate and postgraduate level [25]. Both works resulted in the conclusion that a city planning simulation game can be beneficial for urban planners and general public opinion because it helps people to better understand the problem and they suggested benefits and restrictions.

The participation (co-creation and co-evaluation) of the general public in urban planning processing via virtual reality [26,27], augmented reality [28] and/or mixed reality [29] game and application development is a recently introduced concept. The work of Fares et al. [30] proposes the usage of virtual reality tools and the Internet for the integration of citizens in the decision making process in the planning process. They used software named Vizard to enhance the participation and test the effectiveness of their methodology in Beit Hanoun (Gaza, Palestine) city, which lacks the tools of participation. E-participation using augmented reality technology is discussed in the work of Fegert et al. [31], where they used augmented reality technology for civil participation processes for public construction projects. Wolf et al. [32] investigated how mixed reality can inspire citizens to participate in the early stages of the planning phase (e.g., idea discussion and solution design), rather than later stages (e.g., interventions in the real environment).

The work of Kavouras et al. [33] describes an architectural planning methodology dependent on game engines and 3D software and tools. A digital twin model of an existing warehouse was created using photogrammetry, which in a second phase was imported to a game engine. A simulation video game was developed based on the digital twin model, where the player could spawn basic architecture components (e.g., storage furniture). The architecture planning was achieved by playing the game and placing the components on desired locations and rotations.

Currently, in the field of research into video game technologies in urban planning, there are few papers focusing on actual urban planning intervention case studies. However, similar tools have been applied in other reconstruction areas. An example is the study [34], where game engines are used for architectural historical reconstruction. Raghothama et al. [35] developed transportation simulation models using Unity game engine for the cities of Stockholm and Paris to reduce congestion in traffic and public transport (Stockholm case study) and manage crowds and security during large events in two stadiums (Paris case study). Forssen et al. [36] used Unreal Engine to develop a tool for simplifying the planning process from the early stages and help urban planners increase their knowledge about the urban sound environment and its effects.

Greenwood et al. [37] developed a game-based software application for visualizing digital terrain models and incorporating buildings and vegetation. Additionally, their application includes an established landscape architecture studio as a response to the needs of industry. Vemuri et al. [38] developed a serious game named YouPlaceIt!, where the players take on relevant roles mirroring real-life stake-holders and aim to achieve a consensus related to urban planning issues.

Finally, the work of Xiang and Ong Guo [39] analyzes specific case studies over the last decade to understand how participatory design has been utilized in urban projects and attempts to understand the mechanism behind these interactions by presenting actual urban planning examples in the coastline of Manhattan, urban planning participation discussions on Yogyakarta and Semarang, Tirana and other cities. Additionally, they study how Massively Multiplayer Online Games [40] can be facilitated into online multiplayer interactions. Their conclusion is that a coherent research interest and financial incentives to advance the development of such processes is lacking; however, the tools are already mature and constantly improved according to the needs of the gaming industry.

The following table (Table 1) summarizes the provided literature review by categorizing the citations based on their proposed technology and approach of the urban planning solution and participation process.

**Table 1.** A brief overview of the technologies related to urban planning processes.

| Technology | Papers | Discussion | Pros/Cons and Progress Beyond |
|---|---|---|---|
| Digital Twin and Urban Planning | [12,13,15] | Urban digital twins can support urban planners, designers and the general public. They can be used as a collaboration and communication tool and in decision making processes. | Urban digital twins do not include all the information from the physical world and real life. This limitation can be solved by the creation of BIM systems or interactive 3D platform environments (e.g., gamified reality). |
| Virtual Reality, Augmented Reality or Mixed Reality | [11,27,30–32] | Virtual/augmented/mixed reality tools can inspire the general public to participate in the urban planning process, without restricting any participant with a particular place and time to carry out the process of participation. | Virtual/augmented/mixed reality tools are easy to use and provide usefulness for the participatory and decision making process in urban planning. These technologies matured recently along with hardware and software and further research is needed in this field. Some advantages of this technology can be summarized as follows: (a) improvement in comprehensibility, (b) improvement in the planning process itself, (c) support of interactivity, also for collaborative scenarios, and (d) increased traceability of the planning process. |

**Table 1.** *Cont.*

| Gamified Urban Planning and Participation | [14,16–21,23,33–40] | Gamification of the urban planning process can enhance the participation of the general public in the planning process. Additionally, the interactive environment provided by the game platform makes the urban planning problem more understandable than other approaches. | Using a game environment for the urban planning processes, facilitators and planners can provide a better understanding of the complexity of the problem at hand, which helps in group discussion and decision making processes. A game environment can combine multiple technologies (e.g., virtual reality, computer vision, machine learning, etc.) to enrich the experience of the participants. |
|---|---|---|---|

*Our Contribution*

The contribution of this work can be summarized as follows:

- It reduces the time and cost of the planning process by using online resources for the creation of the digital twin on the best possible quality, depending on the available data.
- It provides a high quality and interactive planning solution, developed on a game engine environment, which provides virtual reality, augmented reality and mixed reality features.
- The produced result can inspire co-creation in the urban planning process. Online or offline distribution sources can be used for sharing the solution(s) to the general public from the early stages of the urban planning process, providing dynamic urban planning capability.
- The created simulation game can inspire participation during the evaluation phase (co-evaluation), which is important for enhancing the democratization of decision making during the urban planning process.

**3. Urban Planning and New Technology**

Urban planning is a multidimensional problem, which uses technical and political processes for solving the problem of design and development over the built environment. In recent years, the urban planning problem has been focused on improving the health and well-being of city residents [41]. Urban green spaces can improve the mental health and well-being of citizens and reduce stress and anxiety [42]. In particular, for older people the urban green spaces are important for health and well-being, because they provide spaces for physical activity and social interaction [43].

Co-creation in urban planning processes via participation in the decision making phases has been widely accepted; however, the challenge to achieve it still remains. According to the work of Kahila Tani et al. [44] planners lack the knowledge of usable tools to reach broader groups of participants and they identified three challenges for the participation problem, namely: (a) effective arrangements of public participation; (b) ability to reach a broad spectrum of people; and (c) the production of high quality and versatile knowledge. They conclude with the need for a variety of participation technologies for encompassing various planning interests among the general public and avoiding encouraging elitist-based participation focusing on those who are able and willing to use power over others. Such technologies may include geographic information systems (GIS) [45], machine learning [46], 3D software [47] or game engines [48].

A GIS database [49] includes various geospatial data such as Digital Elevation Models (DEM) or Digital Terrain Models (DTM) which can be combined with raster images to produce realistic 3D geospatial visualization [50]. Figure 1 illustrates a visual example of this combination, generated in a 3D software. The Figure 1a layer is the raster image texture, the Figure 1b layer is the DTM and Figure 1c is the combination of the previous layers, which corresponds to the final DTM. Some online available sources for the creation of such models are OpenTopography, ArcGIS Satellite, Google Earth, OpenStreetMap, Cesium, Mapbox, etc. Usually, these sources include additional data [51] such as 3D buildings,

vegetation, etc. Figure 2 depicts the addition of 3D buildings in the depicted DTM of Figure 2. The 3D buildings in this case are provided from OpenStreetMap. The final 3D representation is called digital twin [52] and can be used as a starting point for the urban planning processes.

The game engines are powerful tools, which combines 3D software technology, programming interfaces and audiovisual (cinematic) components [53]. In urban planning the game engines can be used for visualizing and navigating in a previously generated digital twin [54]. Additionally, a game engine includes high quality real-time rendering systems combined with high quality light features. Machine learning, artificial intelligence, virtual reality and augmented reality technologies are, also, supported from the game engines [55]. The utilization of these technologies is able to produce an interactive game with the urban planning solution, which can be uploaded on a server and be accessible to everyone.

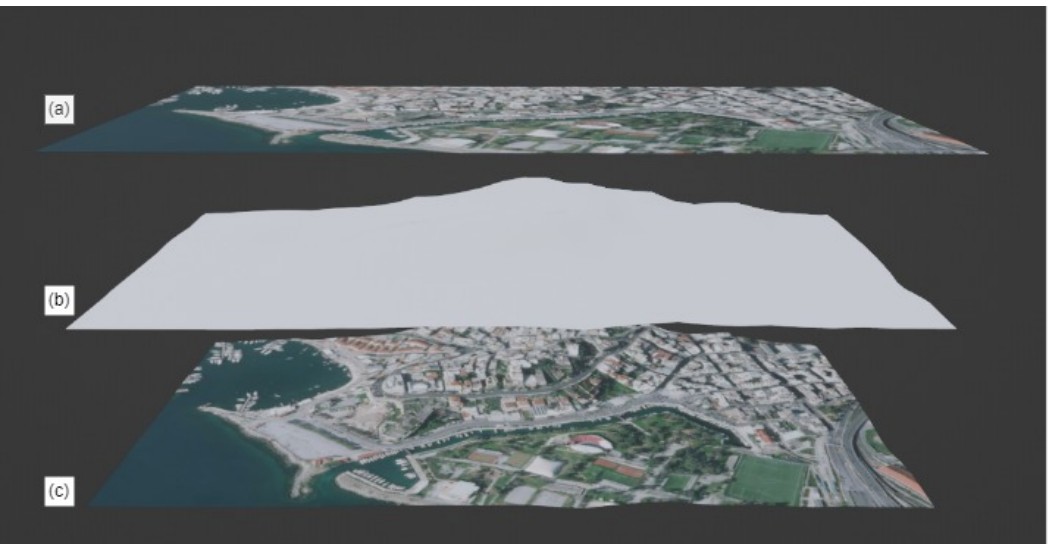

**Figure 1.** A visualized representation of the combination between a DTM and a raster image. (**a**) The raster image texture, (**b**) The 3D Digital Terrain Model and (**c**) The merged 3D reality model created by the combination of (**a**,**b**).

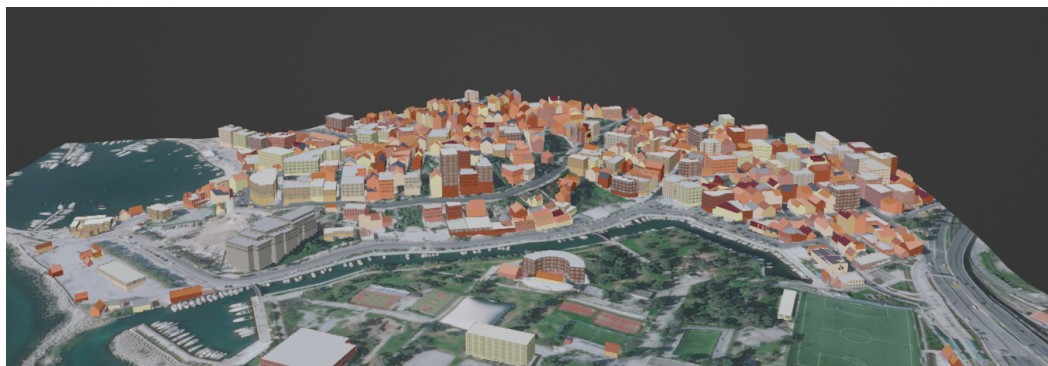

**Figure 2.** Three-dimensional building addition on the DTM of Figure 1.

These technologies are in most cases free, which significantly minimizes the cost of the planning process. In addition, the online geospatial data can be used as is or with minor editing; thus, in a short time period a digital twin can be created. However, these technologies are not free of limitations. Some such limitations could be as follows: (a) low resolution geospatial data that leads to poor quality digital twin, (b) server or networking restrictions or (c) some people may not be able to use them (i.e., old people). These limitations, however, can be overcome by properly designing the overall workflow process

(e.g., using an appropriate dataset depending on the urban planning project needs, selecting a server with higher capacity, organizing conferences/workshops for people who cannot play the game, etc.).

## 4. Proposed Methodology

The proposed methodology aims to facilitate urban planners, architects, designers (experts) and various citizens' groups (non-experts) to design the built environment, considering the principles of nature-based solutions and evaluating each of them, emphasizing minimizing as much as possible the needed time and cost. This can be achieved by using free and open-source data, software and tools, before (e.g., digital twin creation) and during the planning process, as well as for enhancing co-creativity. The proposed solutions can be further packed as an interactive simulation game and distributed (e.g., shared) between groups of experts and non-experts, for co-evaluation. Figure 3 illustrates the components of the proposed methodology, which is divided into four main layers/phases: (a) the data layer, (b) the design layer, (c) the co-creation urban planning layer (game layer) and (d) the experimental test bed layer (pilot layer), described in the following paragraphs.

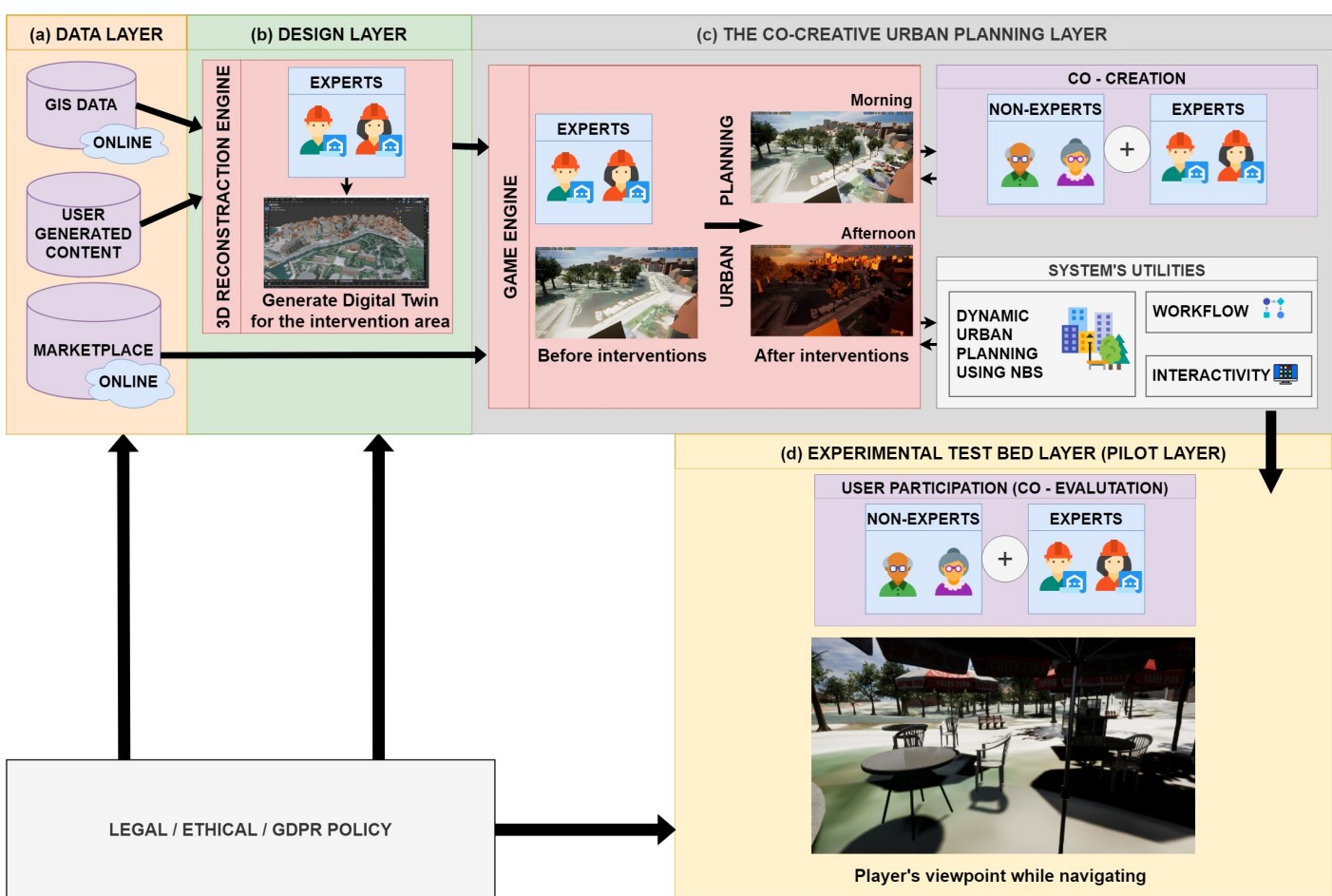

**Figure 3.** Architecture of proposed methodology. The Data Layer (**a**) provides the online and offline (user-generated) data sources (data lake) that will be used at the design and co-creation urban planning layers. In the Design Layer (**b**) the 3D reconstruction engine combines multiple online/offline sources (interconnected systems) for the reconstruction of the digital twin of the area of interest. The Co-Creation Urban Planning Layer (**c**) includes the game engine tools that will be used for the development of the urban planning solution, including co-creation capabilities. Finally, the Experimental Test Bed Layer (pilot layer) (**d**) the urban planning solution is shared to the general public to be evaluated.

The Data Layer represents the available data web services and providers. For our proposed methodology, we need two main kinds of data: (a) topographic data and (b) 3D features for urban planning. Topographic data can be obtained from online GIS web services, such as government websites (e.g., national cadastre, etc.), worldwide GIS services (e.g., satellite data, digital elevation model, etc.). Three-dimensional features can be obtained either by using an online marketplace or they can be created using 3D graphics software. The online marketplace is usually provided by the creator of the game engine.

The Design Layer includes the processes required for the urban planning. The first stage of the urban planning is the creation of a digital twin for the area of interest, by the project coordination expert's team, using the GIS data and other user-generated data, obtained from the data layer. When the digital twin is ready, it is imported to the game engine for the actual planning process. The planning solution(s) provided by the expert team (e.g., planners, architects, engineers) is exported as a simulation game (game layer) and can contain virtual reality, augmented reality or mixed reality capabilities.

The Co-Creation Urban Planning Layer is a component of the design engine. This layer enhances the co-creativity between the experts and non-experts, before and during the urban planning evaluation process. In this phase, the team of experts ask the general public (i.e., non-experts) for opinions and ideas (e.g., what the citizens want). General public participation enhances the co-creativity of the urban planning process, democratizes the proposed solution(s) and provides dynamic urban planning capability and interactivity to the whole process.

The Experimental Test Bed Layer (pilot layer) is the final step of our methodology. In this phase, the developed game is distributed in various groups of experts and non-experts for co-evaluating the solution(s). The distribution can be physical (e.g., USB, CD, live presentation) or online. The online distribution needs a server to host the game platform, while the offline distribution can be achieved with demonstration in conferences and workshops or by mailing physically the solution using a CD or USB media. Depending on the distribution approach, participation constraints may be involved. An example could be the capacity of the maximum players, which can connect at the same time in the game, when it is distributed online. However, such constraints can be easily bypassed in the decision making of the previous layers (e.g., appropriate server selection, age restriction if needed, legal, ethics and GDPR policy handling, etc.). The screening system (e.g., logging in to the platform) can also be applied in the online distribution for minimizing multiple feedback submissions from the same source. In the assessment layer the experts can collect metrics and intelligence for training machine learning algorithms for evaluating solutions.

During the phases of the proposed methodology, non-experts' participation and opinions are provided. The data collection needs to be compatible with the current legislation of the country where the area of interest is. Thus, in our proposed methodology architecture (Figure 3) we include, also, the Legal, Ethics and General Data Protection Regulation (GDPR) Policy, which is mandatory in data collection. This procedure is time consuming and appropriate preparation is needed.

## 5. Data Collection and Design

### 5.1. The Data Lake Interface

The data lake represents the available online and offline resources, where the expert group can use for solving the urban planning problem. Figure 4 illustrates the resources used in the proposed methodology in the different processes. The available Online GIS Data includes cadastre data, digital terrain models (DTM), digital elevation models (DEM), satellite data (remote sensing), etc., which can be used for landscape creation, 3D reconstruction of reality and digital twin creation. The User-Generated Data can be 3D graphics that can be used later in the urban planning phase (e.g., trees, benches, architectural design buildings, etc.), animated rigged characters (e.g., humans, animals, etc.) and rigged objects (e.g., wind power generators, etc.) that can be animated in a second phase for interactivity enhancing (e.g., citizens moving on the scene, birds flying over trees, wind generators

producing power, etc.). The Online Marketplace corresponds to the data lake connected with the game engine software or any other software used in the proposed methodology. In this marketplace, users can find already created 3D meshes, object materials, code for the engine, animated character or objects, etc.

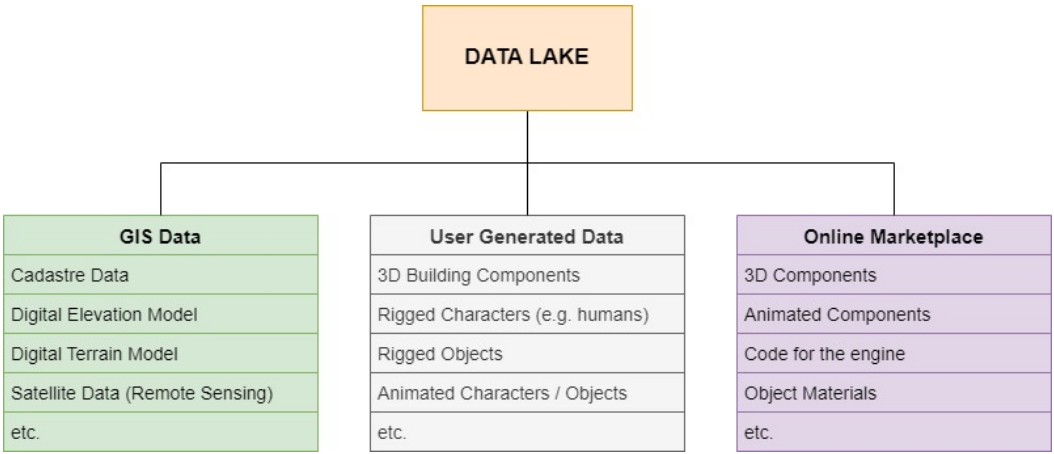

**Figure 4.** The data resources used in the proposed methodology.

### 5.2. The 3D Reconstruction Engine

The 3D reconstruction engine combines a 3D graphic software and its tools with available online data and/or user-generated content for the creation of a 3D reality representation (e.g., digital twin) at an appropriate resolution, according to the needs of the project. Figure 5 illustrated the workflow, which is proposed in this manuscript, for the creation of a digital twin using Blender [56] as the 3D graphic software and a tool (add-on) named Blender-OSM [57]. This is a three step process: (a) terrain creation; (b) terrain material addition; and (c) addition of features (e.g., buildings).

In Figure 5, the terrain creation uses the worldwide 30 meter resolution digital elevation model (DEM) and digital terrain model (DTM) provided from the Open Topography [58]. For the material of the terrain model, the satellite layer of ESRI's ArcGIS satellite layer [59] is used and the buildings are imported from OpenStreetMap [60]. These online GIS data sources are combined in the Blender-OSM add-on, along with many other options of data sources. The expert user (e.g., planner) can further edit the created digital twin before exporting it as an appropriate format (e.g., FBX) for importing it into a game engine. This approach resulted in a digital twin creation with zero cost and under half a day, depending on the size of the area and the further needed editing.

A more traditional approach to the digital twin creation problem would be the usage of digital images and photogrammetry. In this case, appropriate equipment is needed (e.g., drone, DSLR camera, etc.) along with some software to solve photogrammetry problems like the structure from motion (SFM) [61] algorithm for the 3D creation. Depending on the collected dataset, the resultant 3D model may need further clean up or another on-site visit for further image capturing, in the cases where holes exist in the model. A different approach could use a 3D laser scanner and appropriate software [62]. These approaches create digital twins with better resolution and accuracy; however, they need special educated engineers (e.g., knowledge of photogrammetry, UAV pilots, etc.), equipment and compatible software. These requirements increase the cost of the digital twin production and the production time can be from days to months, depending on the size of the area, the weather conditions, etc. [63].

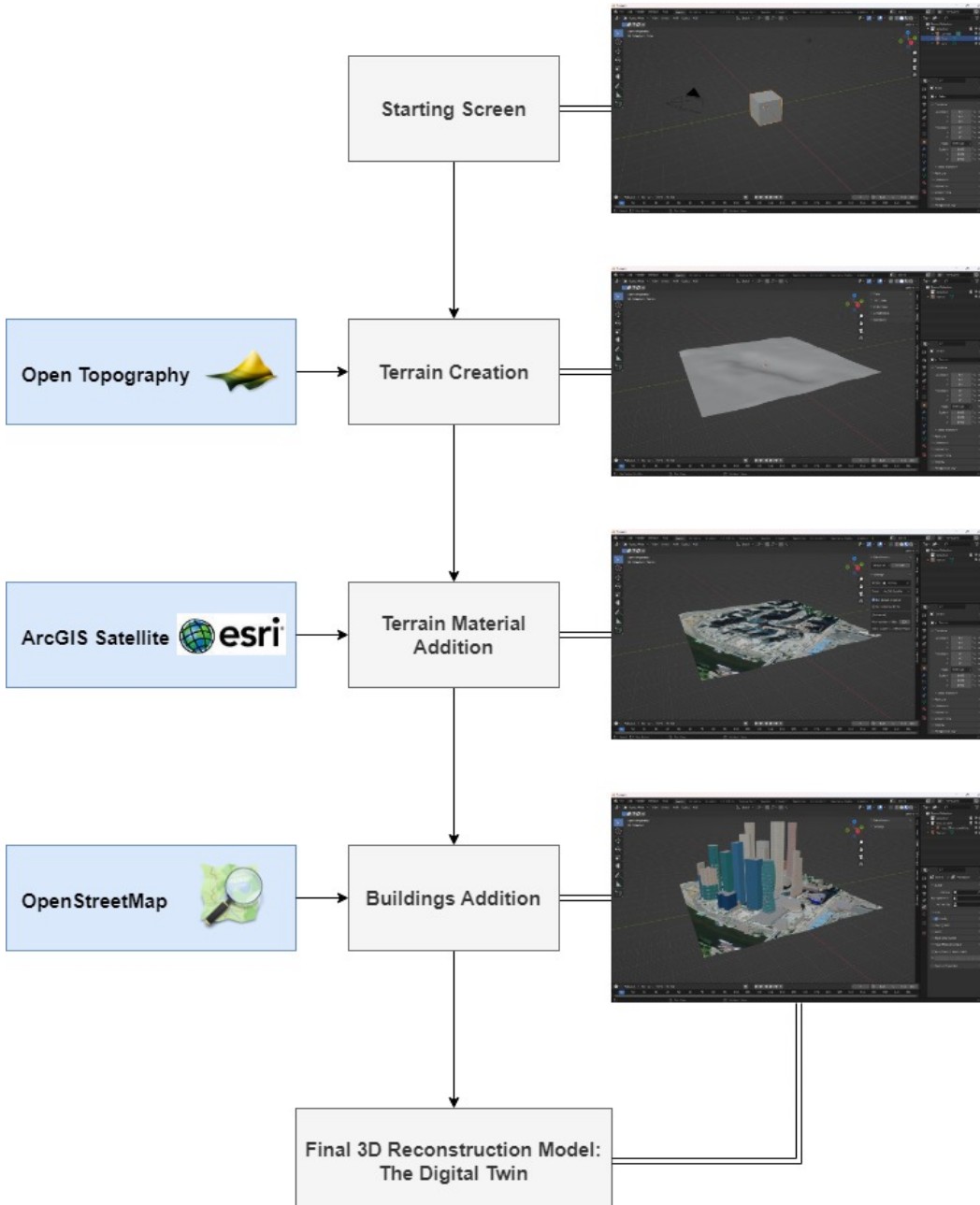

**Figure 5.** The workflow of the proposed 3D Reconstruction Engine.

## 6. The Co-Creative Urban Planning

The proposed methodology described in this manuscript suggests co-creation in the planning process. This is achieved in a two step process, as illustrated in Figure 6. The first step is the design of a game-based urban planning solution by the expert group inside the Game Engine. The second step is the co-creation, where the game-based urban planning solution is shared with other groups of experts and non-experts.

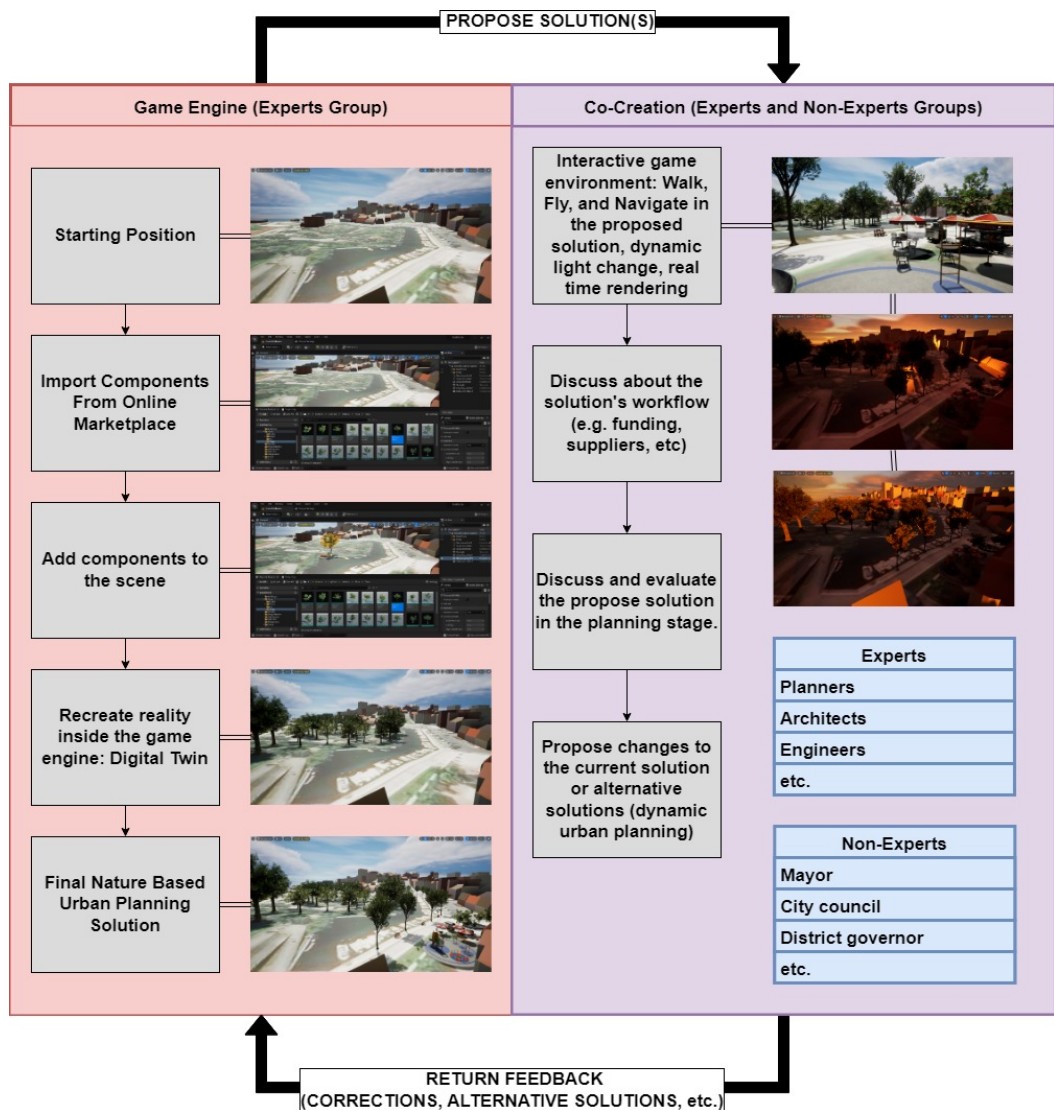

**Figure 6.** Game-based urban planning.

### 6.1. Game-Based Urban Planning

The urban planning process takes place in the game engine environment. In Figure 6, the red box (left) illustrates the workflow of the urban planning. As a first step, the digital twin and other user-generated content are importing to the game engine and combined for synthesizing the starting position or the urban planning process. Next, any additional content (e.g., reusable code, 3D meshes, animated character and/or objects, etc.) is importing from the online marketplace. By adding objects and components in the scene, the digital twin is finalized, representing the reality as much as possible. Using the same tools and objects (e.g., trees, benches, fences, small building, etc.), the urban planning solution is designed by positioning, rotating and scaling the content components to the scent. Usually, the game engines provide, additionally, a game environment, where the users can play the game. Using this feature, the planners can navigate inside their solution and evaluate it. The "design–navigate–evaluate" steps can be repeated as many times needed until the final solution is achieved.

### 6.2. The Co-Creative Functionalities

The co-creation process between the groups of experts and non-experts begins after the finalization of the digital twin and is packaged as an executable file (playable demo solution). The demo solution can be shared to other groups of experts and non-experts

either online via a portal or offline using a medium such as a USB or a CD. Figure 6 illustrates in the purple box (right) a co-creation example. In this example, the simulation game provides a navigation environment where the player can walk, fly and dynamically change the sun position, providing that way a better visualization experience for the general public.

In this early decision making stage, the distribution of the proposed solution can be shared either to small interested groups such as local authorities (e.g., major, city council, local citizens) or the solutions can be shared open to the general public. Such groups can be experts such as urban planners, architects, engineers, etc., and the non-experts such as the mayor, the city council, the district governor, citizens, etc. In this stage of the urban planning, environmental, economical, cultural and other aspects corresponding with the proposed solution can be discussed. Thus, multiple groups of experts and non-experts are cooperating by sharing their opinions and ideas in a co-evaluation process.

The ideas discussed during the co-evaluation stage return as feedback to the urban planners, who work over the area of interest. The feedback may include corrections, alternative solutions or any kind of information or opinion discussed during the co-evaluation process. The urban planners correct their proposed solution, according to the feedback, and propose a new solution. The "creation–evaluation" cycle can be repeated until a solution accepted by the majority of the active groups is achieved. Thus, this process democratizes and decentralizes the final urban planning solution, because it is heavily dependent on the opinion of multiple interested and different expert and non-expert groups, with any academic background.

## 7. Experimental Test Bed

The proposed methodology and the following case studies have been developed under the European Union project euPOLIS "Integrated NBS-based Urban Planning Methodology for Enhancing the Health and Well-being of Citizens: the euPOLIS Approach" (grant agreement No. 869448) [64]. The presented case studies refer to actual areas of interest under the euPOLIS project. The solutions of this paper have been demonstrated for testing the proposed methodology and involved the cooperation of 25 people of the age range 20–65. The experts' team involved the authors of this manuscript, as well as architects and urban planners from the demonstrated areas. The non-expert group included people from the close environment of the expert group (e.g., family members, friends, other work collaborators, etc.). Additionally, our proposed methodology considers people who are unable to access the game platform (i.e., old people). These people can participate in the urban planning process by attending conferences and workshops, as well as by sending them samples of the solution (i.e., video, images, brochures) in digital (i.e., e-mail) or physical (i.e., by mail then a CD, USB or brochures) forms.

### 7.1. Case Study 1: Dilaveri Coast

In this scenario, the area of interest is Dilaveri Coast, in the city of Piraeus (Greece). Figure 7 depicts a preview of Dilaveri Coast from Google Earth Pro, with the area of interest enclosed with the blue polygon. The main characteristic of this coastal area is that it is divided by an artificial canal into the Urban Side (west) and the Green Park Side (east). It is easily observed that the Urban Side suffers from low green areas, which makes Dilaveri Coast a good example for urban planning using nature-based solutions, which aims to improve the health and well-being of the citizens or visitors of the area.

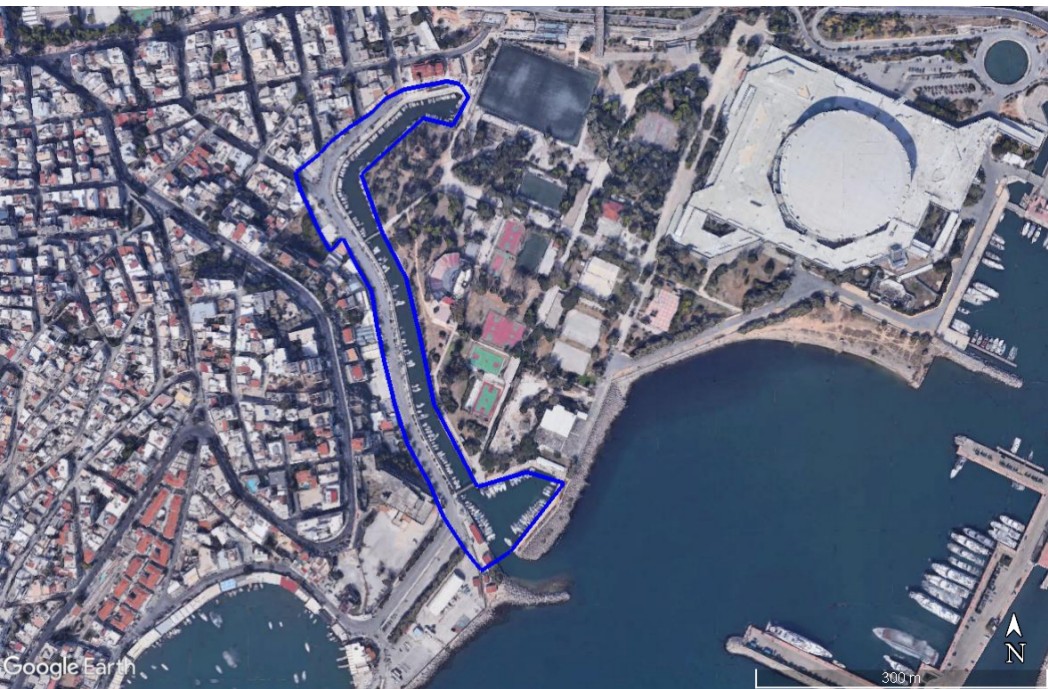

**Figure 7.** View of Dilaveri Coast from Google Earth Pro.

The investigated scenario involves the placement of trees and other features (e.g., benches, playgrounds, etc.) along the shore line and the neighbouring squares in Dilaveri Coast. The described area in Piraeus spans an area of approximately 32 acres on the one side of the canal along the pedestrian sideways. The solution needs to follow NBS instructions provided by sponsors of the project, which means they must focus on being environmentally friendly and on improving health and well-being.

### 7.1.1. The 3D Reconstruction of Reality

For the Reconstruction of Reality we used Blender [56], which is a free and open-source software for implementing 3D graphics and an add-on called Blender-OSM [57]. This add-on downloads and imports real-world terrain data (30 meter resolution); i.e., it imports buildings, roads, paths, trees, river, lakes forests, vegetation and railways from OpenStreetMap. If terrain is provided, these are projected onto the terrain automatically (georeference). With the premium version of this plugin, material can also be added to the buildings.

Figure 8 depicts the reconstructed 3D model of Dilaveri Coast inside Blender, using the Blender-OSM add-on. The accuracy of the terrain model is 30 m, which is provided by the OpenTopography [58] website and covers the entire world. The terrain overlay and buildings are imported from the ArcGIS satellite and OpenStreetMap, respectively. In this stage, we merged the building layer with the terrain layer and exported the model as an FBX file format, which is supported by the Game Engine.

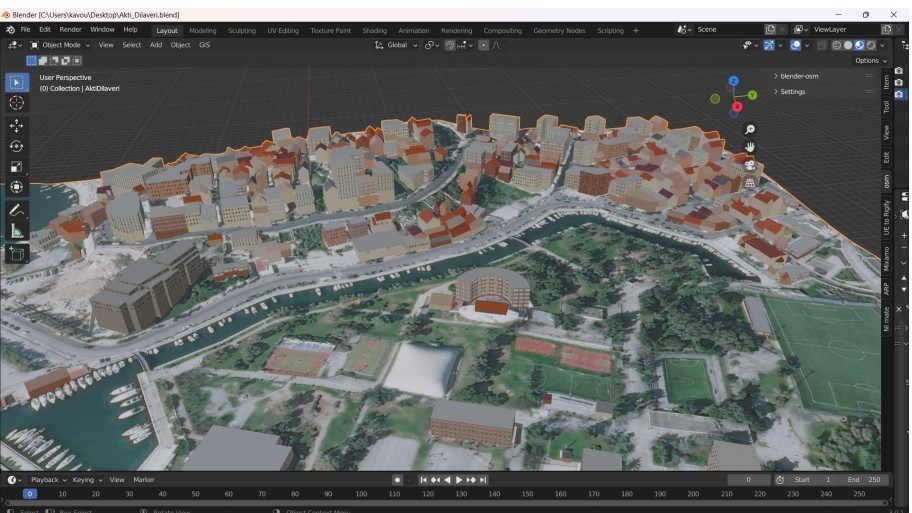

**Figure 8.** Three-dimensional Reconstruction of Dilaveri Coast using Blender and Blender-OSM add-on.

### 7.1.2. The Co-Creative Urban Planning

The actual planning process took place in Unreal Engine 5 (UE5) [65]. UE5 is a free game engine, developed by Epic Games company and released on April 2022. This engine offers a powerful real-time and realistic lighting system, called Lumen, which provides real-time in-game rendering. Furthermore, the engine is based on C/C++, which makes it faster and cross platform. The same capabilities are applied to the games created with it. Furthermore, UE5 supports virtual reality and augmented reality features.

Figure 9 illustrated the digital twin of Dilaveri Coast, when the sun is in the midday position. This corresponds to the reference map, which represent the current reality and will be used as the starting canvas for any designed solution. Figures 10–12 present a proposed NBS with the sun at sunrise, midday and sunset positions, respectively. The designed solution proposes the addition of small trees and benches along the canal. In addition, a playground with some tables, chairs and remote canteens can be added near the church's square, providing that way an area for kids to play, while the parents can be near.

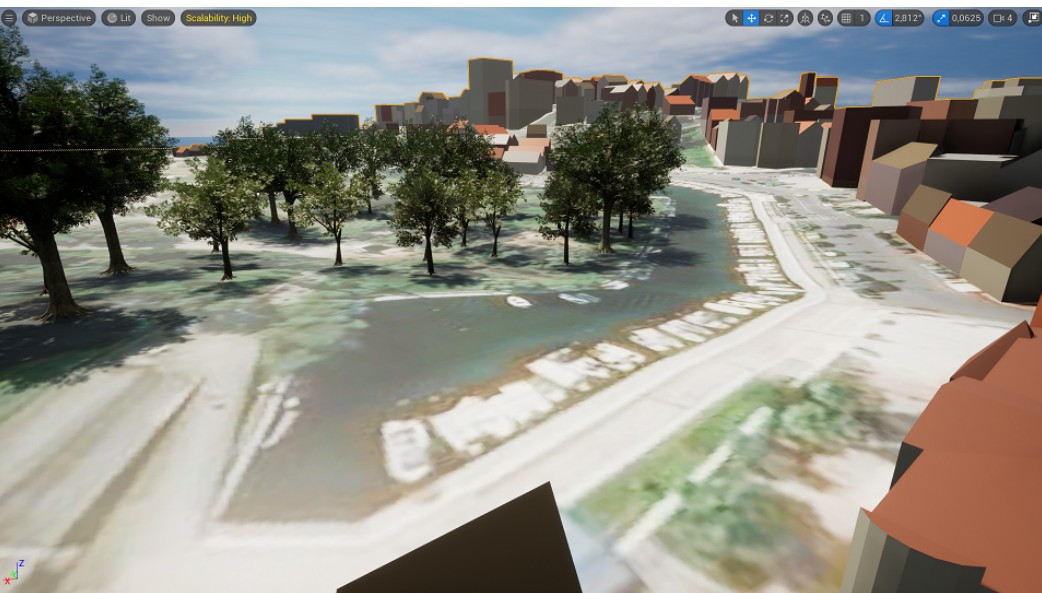

**Figure 9.** The digital twin of Dilaveri Coast inside Unreal Engine. (Starting point for planning).

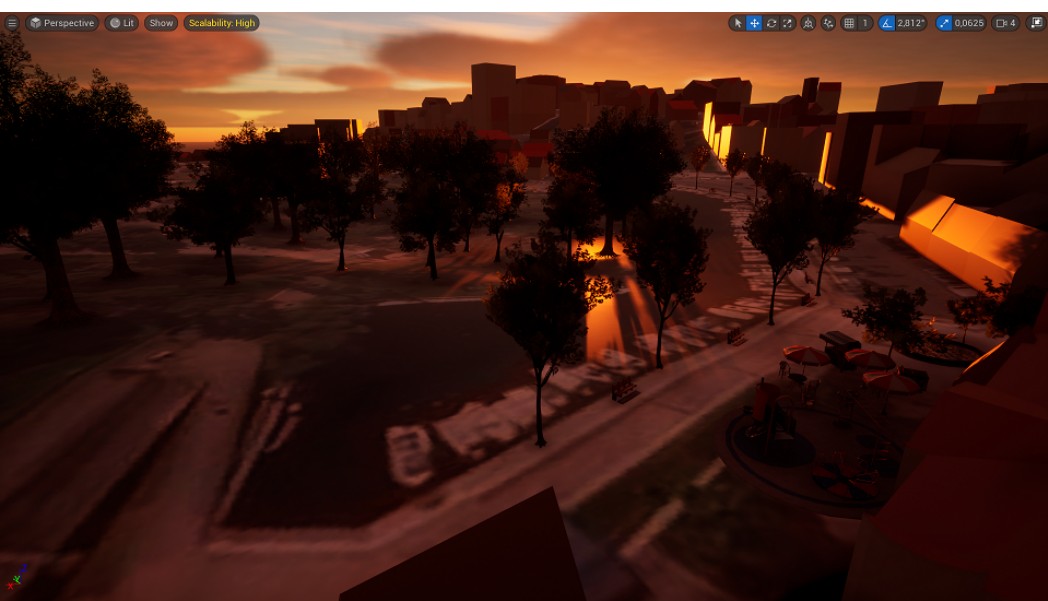

**Figure 10.** A proposed Nature-Based Solution on Dilaveri coast. The sun is at sunrise position.

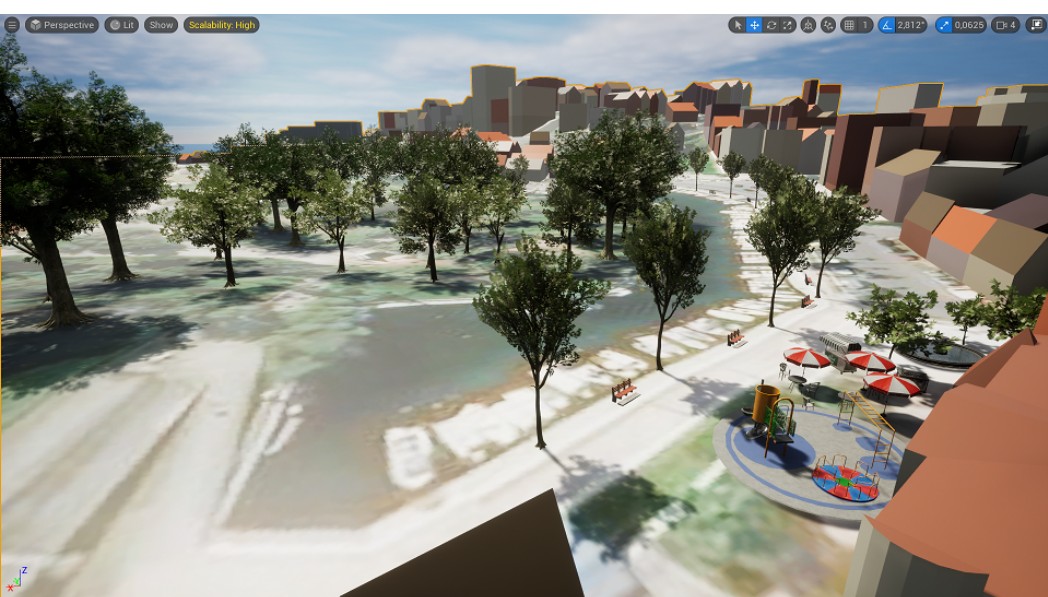

**Figure 11.** A proposed Nature-Based Solution on Dilaveri coast. The sun is approximately at midday position.

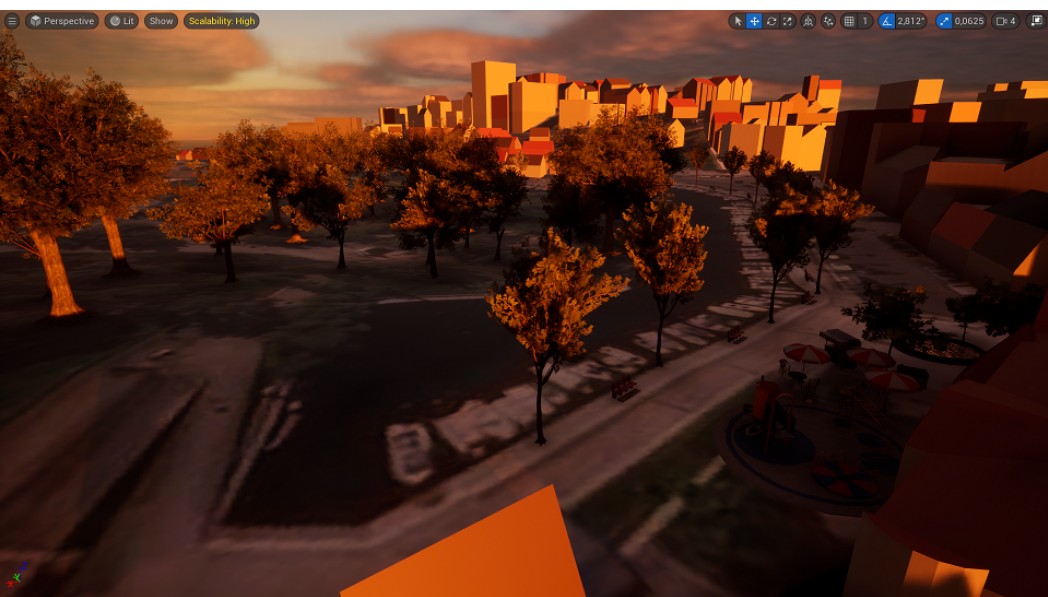

**Figure 12.** A proposed Nature-Based Solution on Dilaveri coast. The sun is at sunset position.

The solution can be evaluated either by playing the game using the available options inside the UE5 editor or by packaging the solution as a standalone executable. In this case study, the evaluation of the solution is carried out by us. Sharing the solution and asking for feedback from the general public or any other group of experts and non-experts requires specific legal approval, which needs preparation and takes time (in our case some months of preparation). Thus, for both the case studies, we evaluated our proposed solutions by playing the game and navigating using the available options of UE5 editor. Figure 13 presents a perspective of a human eye from an angle of a citizen standing under a canteen umbrella. An evaluation comment can be that the shadows in this picture, provided by the addition of the trees and the umbrellas, create a cool environment, away from the sunny areas, which can be unbearable during the summer months. Moreover, the addition of benches along the canal and under the trees provide a place for relaxation.

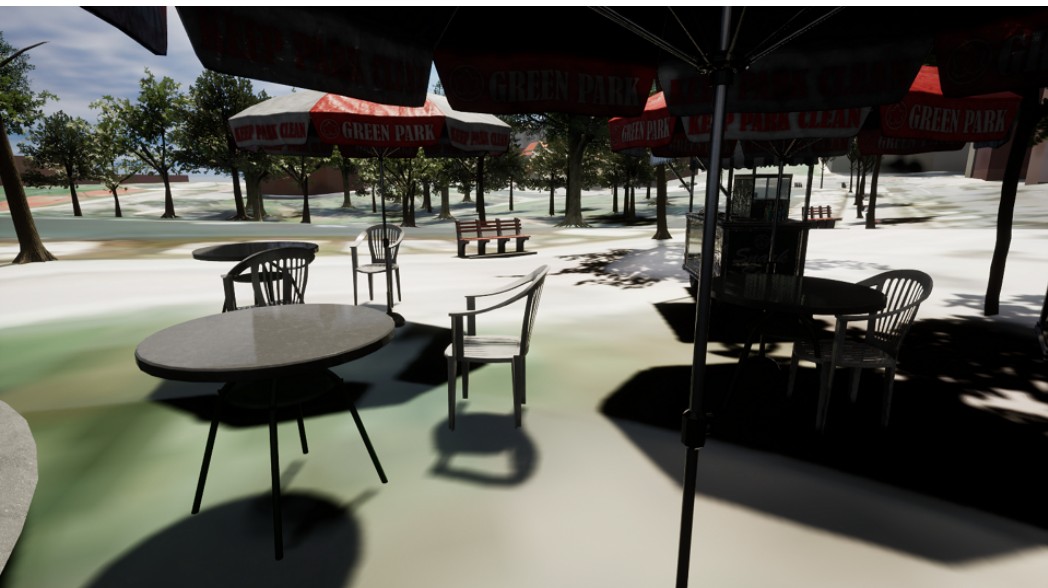

**Figure 13.** A human Eye perspective of our proposed solution.

### 7.2. Case Study 2: Pileparken Social Housing Estate

In this case study, the area of interest is Pileparken 6, a social housing estate, which lacks green spaces. Figure 14 encircles the area of interest in the green polygon. This housing estate was built with public funding; thus, the municipality can dispose of every third vacant dwelling. This complex contains 4 apartment blocks, 117 apartments and 1700 inhabitants. Additionally, it is one of the three districts of the Municipality of Gladsaxe's Social Balance programme. The programme is primarily centered around the municipality's larger social housing districts (and their surroundings), working on a local community level with subjects such as health, safety, citizenship and volunteering and urban living. With The Strategy of Citizenship, the Municipality of Gladsaxe has a strong focus on developing the municipality together with its citizens, which also is a central part of the project in Pileparken. The ambition of the Strategy of Citizenship is that "all citizens take responsibility for and involve themselves in the development in the local communities, for the benefit of the individual as well the society".

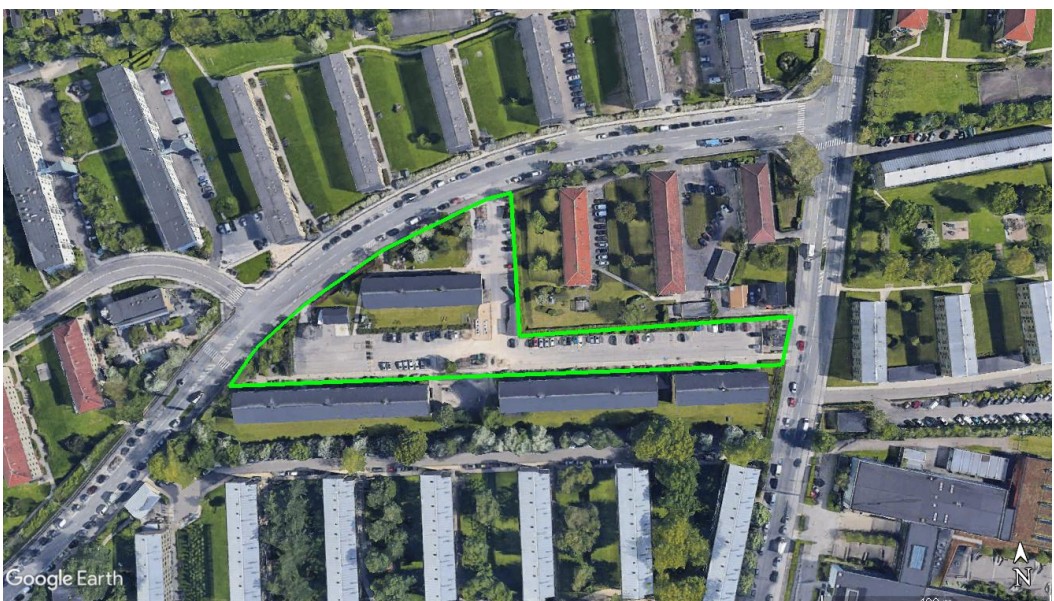

**Figure 14.** Preview of Pileparken social housing estate from Google Earth Pro.

The Pileparken housing estate has three major problems: (a) green areas have very low biodiversity, (b) green areas do not inspire residents to use them and (c) residents do not use the open outside spaces. Additionally, the playground is covered with sand (a permeable material that gets compressed over time). Pedestrian lanes cover all entrances to buildings but not the green open areas. The path design is very rational and creates long straight lines. This creates problems with bicycles and mopeds driving at high speed on the lanes.

Possible solutions could be the creation of more inspiring green spaces, enhancement of biodiversity and creating spaces for outdoor physical activities. Additionally, the improvement of shading areas, reuse of rainwater and its separation from the grey wastewater are of high importance. Moreover, like Dilaveri Coast, the proposed solution needs to be nature-based, environmentally friendly and to improve the health and well-being of the citizens.

### 7.2.1. The 3D Reconstruction of Reality

The first step of our methodology is the reconstruction of the digital twin, which includes the Pileparken housing estate and the surrounding area. Figure 15 depicts the created digital twin using Blender and Blender-OSM. The resolution of the terrain model

was 30 m and was provided by OpenTopography. The buildings were downloaded from OpenStreetMap and the image overlay from ArcGIS satellite.

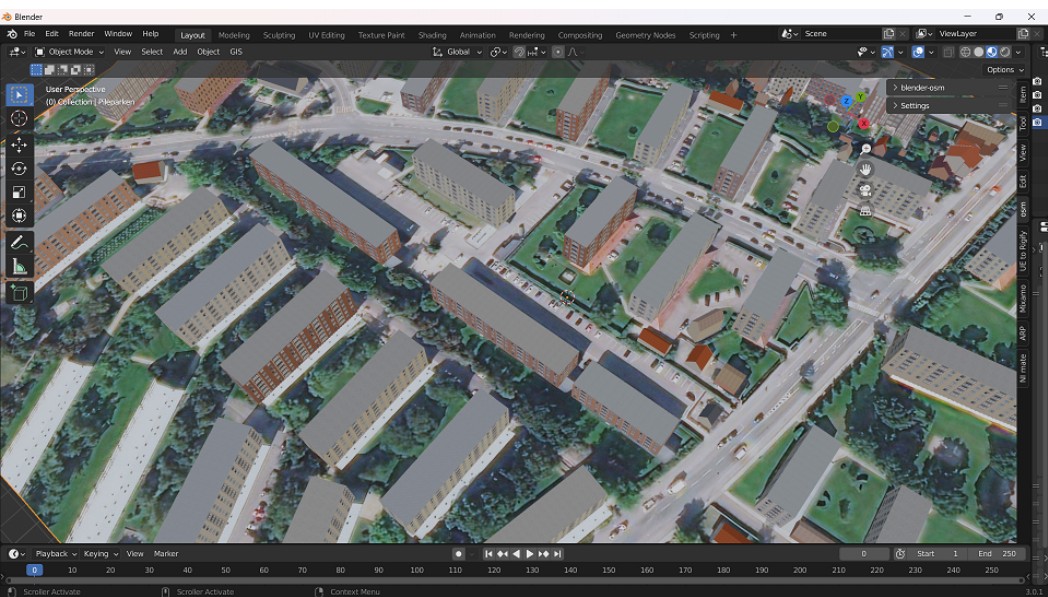

**Figure 15.** Digital twin of Pileparken housing estate and surrounding area.

### 7.2.2. The Co-Creative Urban Planning

The urban planning process takes place inside the game engine (UE5). Figure 16 shows the reference digital twin, which is used as the starting point for designing the alternative NBS. In this manuscript, one of these solutions is presented. Figures 17–19 illustrate the sunrise, approximately at midday sun position and at sunset, respectively. Figure 18 depicts four angles of our proposed solution in the Pileparken housing estate.

One of the main problems of the area was that the playground was covered with sand. A solution for this problem was to create a basketball field for older kids (Figure 18a) and an area for younger kids (Figure 18b). A central plaza with a fountain in the middle, multiple benches and remote canteens separates the playgrounds. In this central area, the residents can walk, sit and relax, solving the problem of the residence not using the outside area by giving them incentives to use the outside area of Pileparken.

Furthermore, the proposed solution covers the park area with trees, which creates shaded areas and increases the biodiversity of the area (Figure 18a,b). However, some of the parking area remained, to provide parking slots for the residences (Figure 18c,d). Trees and bushes (small fiora) have been added in this area, too. That way, the green areas are increased, covering the whole area of interest. Additionally, a WC building (Figure 18b) was added near the entrances/exits of the park.

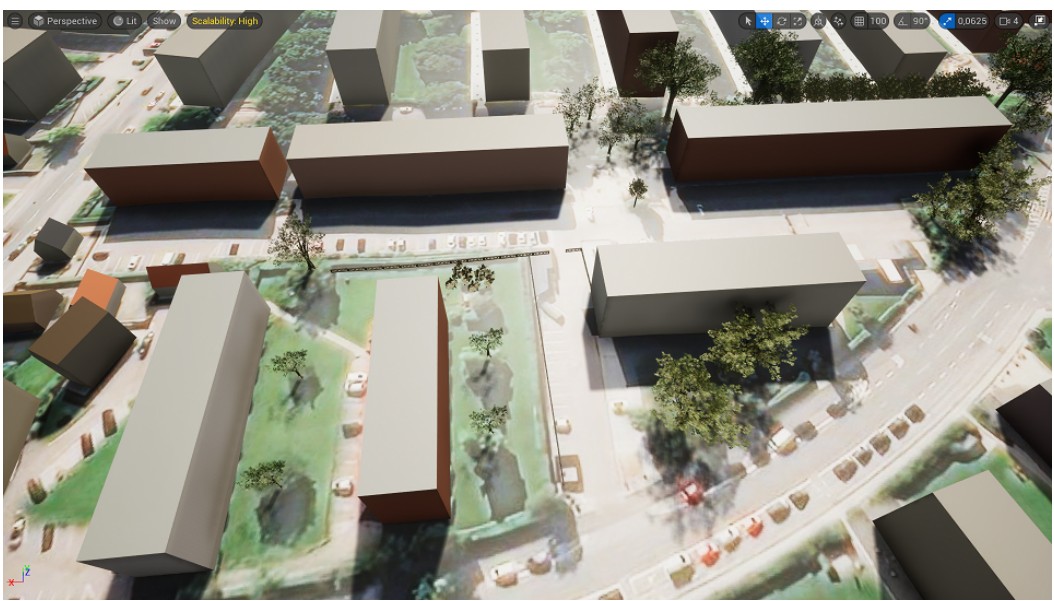

**Figure 16.** The digital twin of Pileparken inside Unreal Engine. (Starting point for planning.)

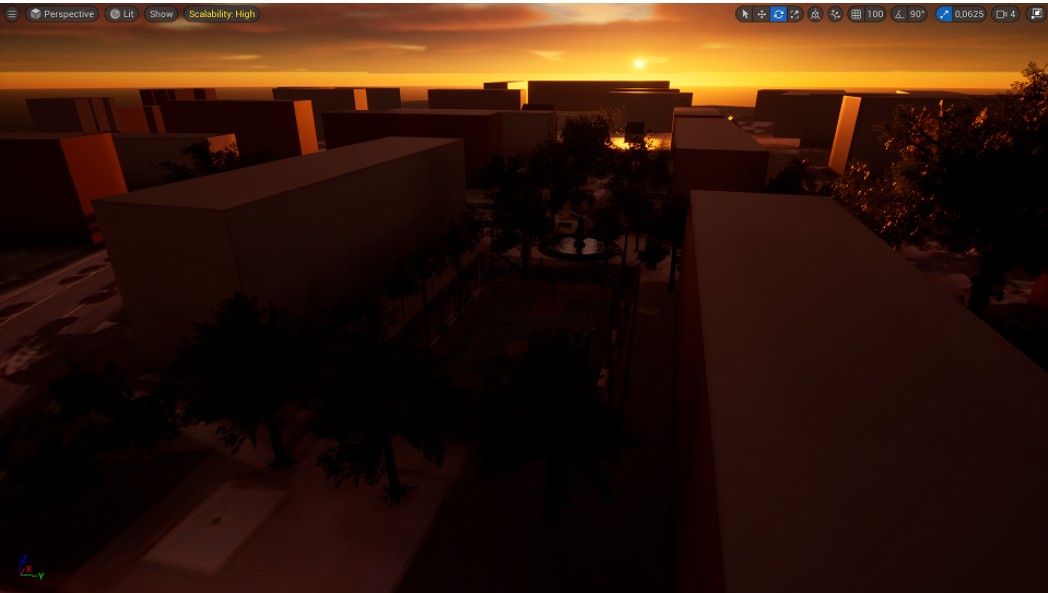

**Figure 17.** A proposed Nature-Based Solution on Pileparken. The sun is at sunrise position.

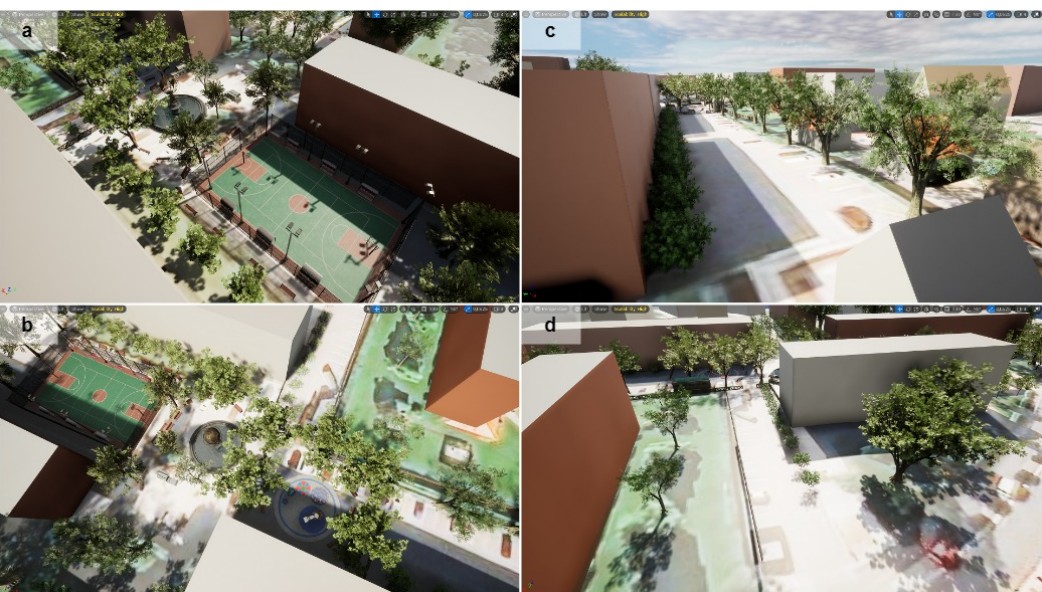

**Figure 18.** A proposed Nature-Based Solution on Pileparken from four different angles: (**a**) basketball view, (**b**) an overall view on the area, (**c**) side parking place and (**d**) top parking place. The sun is approximately at midday position.

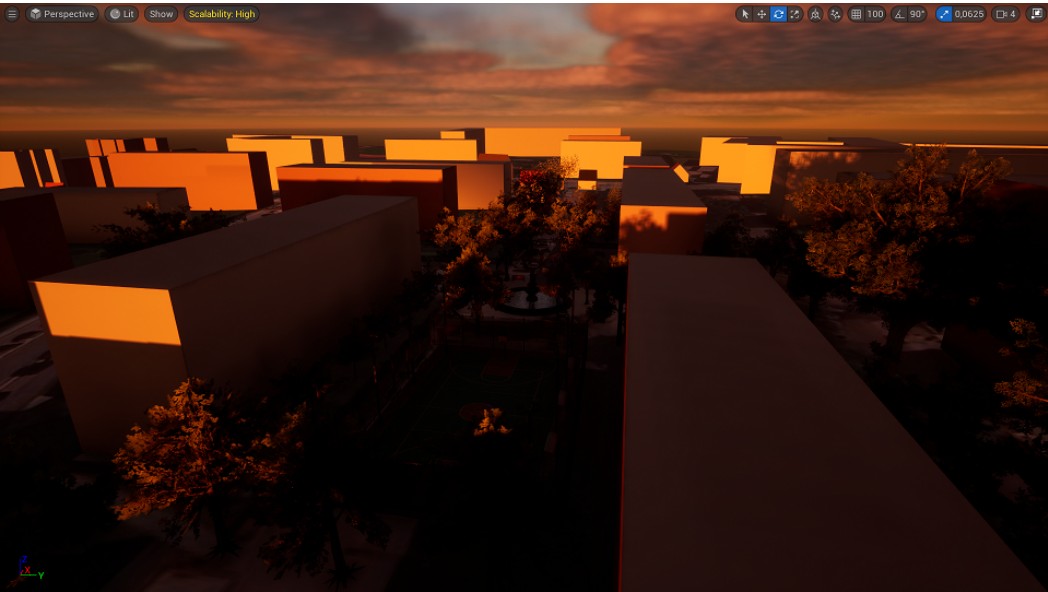

**Figure 19.** A proposed Nature-Based Solution on Pileparken. The sun is at sunset position.

By playing the game using the options provided by the UE5 editor, we evaluated the solution. Figures 20–22 depict three different views from the estimated eye level of an adult. Figure 20 illustrates the view of a first stop of a resident after parking his car on the right side (of the park) parking area. Figure 21 depicts the view of a resident, when he enters the park area, while Figure 22 is the view of a resident near the central park (fountain area) and the basketball field. These three different angles help to estimate how the proposed NBS can be seen and be felt before any intervention in the actual area. In this case, the proposed solution inspires the residents to use the outside area for a variety of activities, which was impossible before.

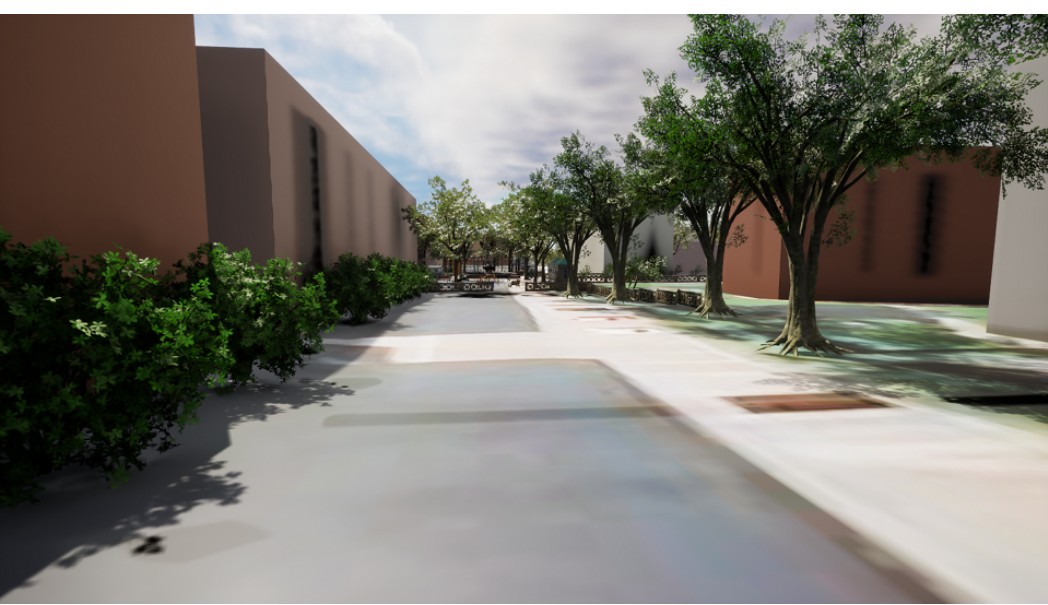

**Figure 20.** View of the right side parking area. We can see that the addition of bushes and trees create a significant difference in green areas, compared to the reference model.

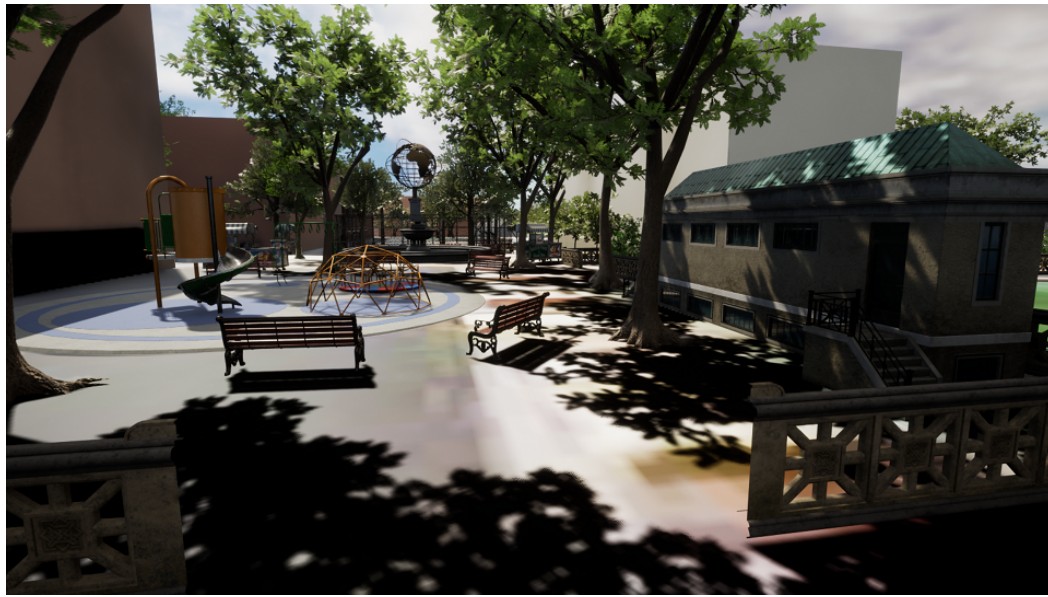

**Figure 21.** View of the park's entrance. The addition of the green areas, fountain, benches and kids' playground can inspire the residents to use the outdoor area. The building on the right is the WC.

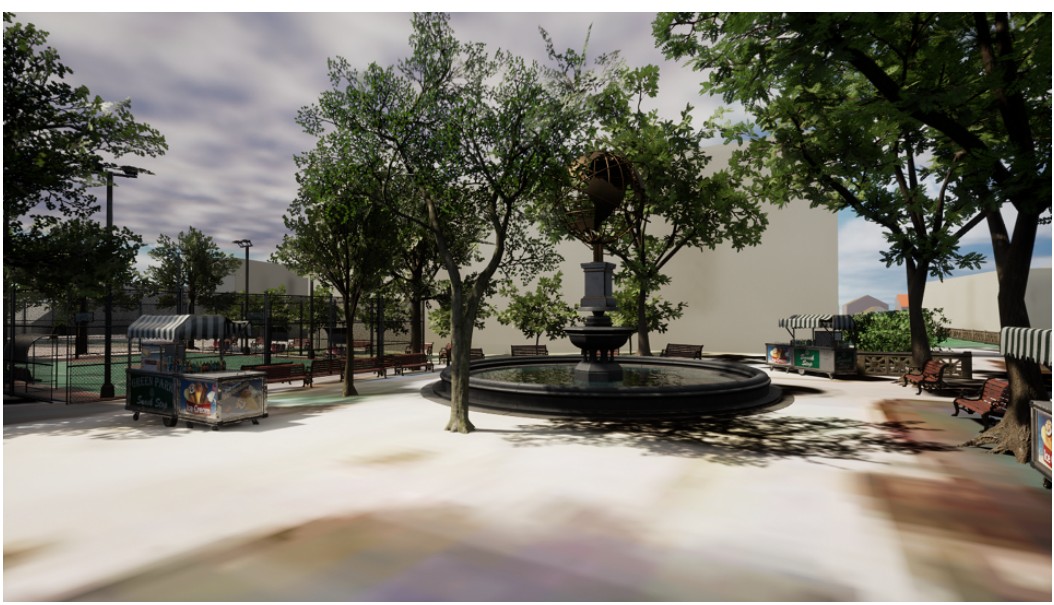

**Figure 22.** View of the park's center area and the basketball field.

## 8. Discussion

This work proposes an urban planning methodology, which minimizes the time and cost of the planning process, Our proposed methodology emphasizes enhancing the co-creation during the planning process, with the aim of democratizing and decentralizing the final solution. Non-expert participation, especially of the residents of the area, is a key component for a successful final solution. Other expert groups can help the urban planners by providing technical feedback or proposing alternative ideas and approaches, during the evaluation phase.

In our methodology, we propose the usage of online data sources for the creation of a 3D representation of reality. Better terrain and building geometries result in better representation of the area after the design. Free online geospatial data may not be as accurate as the ones created using photogrammetry could be. However, this is dependent on the location of the area of interest and the availability of data in that area. For example, high cadastre terrain models may be available, dependent on the country and/or the region, where the area of interest is located.

The design of the planning solution inside a game engine permits experimentation with different kinds of light angles, providing the ability to design, visualize and simulate on a 1:1 scale. In-game real-time rendering may be provided depending on the game engine. Virtual reality, augmented reality and/or mixed reality capabilities are, usually, supported from the game engine and can be included in the gamified solution for enriching the interactivity and attracting more citizens, especially of younger ages, to participate in the co-creation/co-evaluation processes.

Legal, ethics and GDPR policies need to be taken into account during the co-creation/co-evaluation phases. Our proposed methodology suggests the distribution of the gamified solution to the general public either online or offline. Thus, anyone can express opinions and provide feedback of some form. For this matter, personal data need to be collected according to the local legislative system. The legal process may need time, which may delay the whole planning process for days, weeks or months.

The usage of a game engine for the planning process and generally the idea of sharing the planning solution via an interactive game enhance the understanding of the problem. Each interested person (e.g., citizen, resident, visitor, etc.), without any specific educational background, is able to express his/her opinion. The tools provided by the game engine can inspire the planners to create additional features (e.g., non-playable characters walking, driving or interacting in any way with the virtual environment). This is still an open-field for research and possible research could include statistical data from the co-creation and

co-evaluation research or propose innovative ideas, which can help in inspiring the general public's participation during the planning process.

This work has been developed under the European Union project euPOLIS "Integrated NBS-based Urban Planning Methodology for Enhancing the Health and Well-being of Citizens: the euPOLIS Approach" (grant agreement No. 869448) and the proposed methodology has not been efficiently tested yet. However, both of the presented case studies correspond to real urban planning interventions. The results and observations of this research work can be summarized as follows:

- The 3D reconstruction (digital twin) of a real area for urban planning needs can be achieved easily, quickly and at a low-cost using open geospatial data from multiple online sources.
- Using the digital twin and navigating in the area, using the game engine, provided a better understanding of the problem at hand and made the decision making easier and more efficient (i.e., it was easier to estimate the position to plant the trees).
- The real-time interactivity provided by the game engine also helped in the decision making process (i.e., by adding a tree in the scene and moving the sun position, we could see its shading, which helped us later on in the addition of benches, tables, etc.).
- By demonstrating the ideas in non-experts (i.e., family members and friends), by showing them rendered images at first and then by playing the game and navigating in the 3D environment, we observed that they understood better the results when they navigated and experienced the solution. A significant feedback comment was that they "felt" the wind by observing the movement of the trees and the cold and hot areas, dependent on whether or not there was tree shading.

Future work includes the testing of our methodology in workshops, open discussions or using online participation portals, collecting the appropriate statistical data for analyzing and comparing the participation trend of our approach with the state of the art approaches and improving the proposed methodology accordingly. A suggestion concerning how to use the proposed methodology could be the creation of a high-fidelity 3D model in small intervention areas, while on wider areas like our case studies a lower resolution 3D model can be used. This model can be imported to a game engine for the urban planning solution development. The final solution can be shared on the network as a 3D mass multiplayer online game, where the players can rate the solution and propose suggestions in real time. Improvements in the solution can be distributed as game updates. For old people, who cannot connect on the game, workshops and conferences can be organized, where they can participate and express their opinion.

### 9. Conclusions

In this manuscript, an urban planning methodology is presented. The key aspects of the proposed methodology are the minimizing of time and cost of the planning process, while suggesting ways for maximizing the general public's participation during the planning process. The proposed methodology was tested in two case studies and this manuscript's conclusions are as follows:

- The usage of online databases and ICT tools significantly minimizes the time and cost of the 3D reconstruction of the reality (digital twin).
- The usage of a game engine, in the planning and evaluating phases, helps the expert group to better understand what interventions are needed.
- The game engine provides virtual reality, augmented reality and mixed reality features, which can be included in the final solution for enhancing the interactivity and help the non-expert groups to better understand the solution.
- The co-creation, between the groups of experts and non-experts, improves the decision making process during the planning and evaluation phases.

- The co-creation and co-evaluation can provide a democratized and decentralized final solution produced by the cooperation of urban planners working on the solution and the general public.

  Disadvantages of our proposed methodology might be the following: (a) the absence of online high-resolution geospatial data that can be used for the digital twin creation and the accuracy/visualization of the final result is analogous to the provided data (e.g., digital terrain elevation, digital elevation models, online GIS data, etc.); (b) the co-creation and co-evaluation phases need appropriate preparation such as the creation of an online portal, which cooperates with the ethics and local legislative system, that in some cases takes time (i.e., the needed preparation time for the distribution of the case studies, presented in this manuscript, to the general public is months).

**Author Contributions:** Conceptualization, I.K., E.S., E.P., I.R., A.D., and N.D.; methodology, I.K., A.D., and N.D.; writing—original draft preparation, I.K., and E.P.; writing—review and editing, E.S., E.P., I.R., A.D., and N.D.; visualization, I.K. All authors have read and agreed to the published version of the manuscript.

**Funding:** This work is funded by the European Union Funded project euPOLIS "Integrated NBS-based Urban Planning Methodology for Enhancing the Health and Well-being of Citizens: the euPOLIS Approach", under the Horizon 2020 program H2020-EU.3.5.2., grant agreement No. 869448.

**Institutional Review Board Statement:** Not applicable.

**Informed Consent Statement:** Not applicable.

**Data Availability Statement:** Not applicable.

**Acknowledgments:** This work is supported by the European Union Funded project euPOLIS "Integrated NBS-based Urban Planning Methodology for Enhancing the Health and Well-being of Citizens: the euPOLIS Approach", under the Horizon 2020 program H2020-EU.3.5.2., grant agreement No. 869448.

**Conflicts of Interest:** The authors declare no conflict of interest.

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
