# Peer review of "A Low-Cost Gamified Urban Planning Methodology Enhanced with Co-Creation and Participatory Approaches"

_sustainability, doi:10.3390/su15032297_

Round 1

Reviewer 1 Report

This paper presents an interesting theme on game-based urban planning methodology that combines both digital twin and real development of cities. The authors did a nice job to propose a framework of participatory planning and design in virtual society, and show us two cases studies in solving the urban green problem. Therefore, I would like to recommend to accept this paper after some minor improvement:

1.     In the beginning, the authors have provided a good literature review on the usage of game and digital technologies in urban planning, but it would be better if they can further mention to what extend that current gamified urban planning has been employed in real practice, e.g., by showing some references and cases here.

2.     In the section of methodology, the authors provide a nice introduction about the construction of software and dataset, but more information about management of the players is required, for instance, is there a threshold to participant into the game, how about the participatory size, any screening system exist in these processes.

3.     Similarly, more information about the participators in two case studies are expected, i.e., how many people are involved, and who are these experts and non-experts selected or invited, what about these old people who cannot access into game platform.

4.     In the discussion, the authors may mention the potential implications on urban policy and management by this new methodology in the future, and some suggestions to employ this planning method appropriately.  

Author Response

Bellow find out our responses in the attached document.

Reviewer 2 Report

I have no objections to the point, but on the whole article I think that a much more developed theoretical part is needed, the description of the methods, the results and the conclusions are more reasoned.

Author Response

“I have no objections to this point, but on the whole article I think that a much more developed theoretical part is needed, the description of the methods, the results and the conclusions are more reasoned.”

In the revised manuscript we have included a new Section (Section 3), which discusses the theoretical part of urban planning and new technological achievements as reviewer suggests.

Reviewer 3 Report

Please find my comments below:

1- The abstract should be briefly written to decriable the purpose of the research, the principal result, and major conclusions. authors should revise it.

2- authors should add more literature reviews related to the topic of study.

3- Labels all layers in Figure 1.

4- Double check the c numbering in line 159.

5- The results argumentation isn't clear, authors should  discuss it in more detail.

Author Response

(The authors gave the same response as above.)

Round 2

Reviewer 2 Report

I am happy with the changes made.